# Leveraging Future Relationship Reasoning for Vehicle Trajectory Prediction

**Daehee Park[1], Hobin Ryu[2], Yunseo Yang[1], Jegyeong Cho[1], Jiwon Kim[2], Kuk-Jin Yoon[1]**
[1]Korea Advanced Institute of Science and Technology     [2]NAVER LABS
`{bag2824,acorn,j2k0618,kjyoon}@kaist.ac.kr`
`{hobin.ryu,g1.kim}@naverlabs.com`

## Abstract

Understanding the interaction between multiple agents is crucial for realistic vehicle trajectory prediction. Existing methods have attempted to infer the interaction from the observed past trajectories of agents using pooling, attention, or graph-based methods, which rely on a deterministic approach. However, these methods can fail under complex road structures, as they cannot predict various interactions that may occur in the future. In this paper, we propose a novel approach that uses lane information to predict a stochastic future relationship among agents. To obtain a coarse future motion of agents, our method first predicts the probability of lane-level waypoint occupancy of vehicles. We then utilize the temporal probability of passing adjacent lanes for each agent pair, assuming that agents passing adjacent lanes will highly interact. We also model the interaction using a probabilistic distribution, which allows for multiple possible future interactions. The distribution is learned from the posterior distribution of interaction obtained from ground truth future trajectories. We validate our method on popular trajectory prediction datasets: nuScenes and Argoverse. The results show that the proposed method brings remarkable performance gain in prediction accuracy, and achieves state-of-the-art performance in long-term prediction benchmark dataset.

## 1 Introduction

For safe autonomous driving, predicting a vehicle's future trajectory is crucial. Early heuristic prediction models utilized only the past trajectory of the target vehicle ( Lin et al. (2000); Barth & Franke (2008)). However, with the advent of deep learning, more accurate predictions can be made by also considering the vehicle's relationship with the High-Definition (HD) map ( Liang et al. (2020); Zeng et al. (2021)) or surrounding agents ( Lee et al. (2017); Chandra et al. (2020)). Since surrounding vehicles are not stationary, predicting relationships with them is much more complicated and has become essential for realistic trajectory prediction. Furthermore, since individual drivers control each vehicle, their interaction has a stochastic nature.

Previous works modeled interaction from past trajectories of the surrounding vehicles by employing pooling, multi-head attention, or spatio-temporal graph methods. However, we observed that these methods easily fail under complex road structures. For example, Fig. 1 shows the past trajectories of agents (left) and the attention weights among agents (right) obtained by a previous method (Mercat et al. (2020)) that learned the interaction among agents using multi-head attention (MHA). Since agents 0 and 4 are expected to join in the future, the attention weight between them should be high. However, the model predicts a low attention weight between them, highlighting the difficulty of reasoning future relationships between agents based solely on past trajectories. Incorporating the road structure should make the reasoning process much easier.

The decision-making process of human drivers can provide insights on how to model interaction. They first set their goal where they are trying to reach on the map. Next, to infer the interaction with surrounding agents, they roughly infer how the others will behave *in the future*. After that, they infer the interaction with others by inferring how likely the future path of other vehicles will overlap the path set by themselves. The drivers consider interaction more significant the more the future paths of other vehicles overlap with their own. We define the interaction from this process as a "*Future Relationship*". We use the following approaches to model Future Relationship, as shown in Fig. 3.

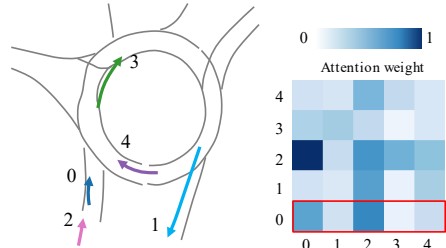

Figure 1: Past trajectories and corresponding attention map between agents from previous work (Mercat et al. (2020)). A weak relationship is inferred between agents that will highly interact in the future: agents 0 and 4.

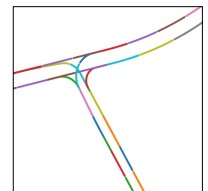

Figure 2: Lane segments represented in different colors.

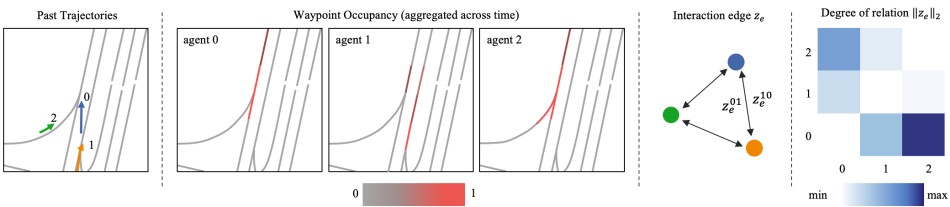

Figure 3: Key concept of the proposed method: From past observed trajectories, we predict the lane that a vehicle will pass in the future. The Interaction between agents is represented by an edge connecting their nodes, and is determined by the probability that two agents will pass adjacent lanes. The greater the probability, the higher the expected interaction.

First, we obtain the rough future motion of all vehicles in the scene. Since vehicles mainly move along lanes, we utilize lane information as strong prior for representing the rough future motion of vehicles. Because lane centerlines contain both positional and directional information, rough future motion can be represented as waypoint occupancy. The waypoint occupancy is defined as the probability of a vehicle passing a specific lane segment at every intermediate timestep. In the middle of Fig. 3, each agent's waypoint occupancy is shown. Here, we aggregated the temporal axis for simplification. The probability that the vehicle passes that lane during the prediction horizon is drawn using the tone of red color.

Second, based on the waypoint occupancy, we infer the Future Relationship in probabilistic distribution. In most vehicle trajectory prediction methods, the interaction between agents is still made in a deterministic manner. However, we take note that interaction between vehicles is highly stochastic, and there can be multiple possible interactions. The deterministic relation inference averages out diverse interactions, interrupting socially-aware trajectory prediction. Therefore, we define Future Relationship in Gaussian Mixture (GM) distribution. Motivated by Neural Relational Inference (NRI) ( Kipf et al. (2018)), we propose a method to train the diverse interaction distribution explicitly.

In summary, our contributions are:

1) We propose a new approach for modeling the interaction between vehicles by incorporating the road structure and defining it as *Future Relationship*.

2) We propose to infer the Future Relationship in probabilistic distribution using Gaussian Mixture (GM) distribution to capture diverse interaction.

3) The proposed method is validated on popular real-world vehicle trajectory datasets: nuScenes and Argoverse. In both datasets, there is a remarkable improvement in prediction performance and state-of-the-art performance is achieved in the long-range prediction dataset, nuScenes.

## 2    RELATED WORK

### 2.1    GOAL-CONDITIONED TRAJECTORY PREDICTION

Predicting future trajectories at once is a challenging task. Instead, goal-conditional prediction, which samples goal candidates and then predicts trajectory conditioned on them, is helpful and has shown

state-of-the-art performance, especially in long-range prediction tasks ( Zhao et al. (2021); Gu et al. (2021); Phan-Minh et al. (2020); Chai et al. (2020); Zhang et al. (2021)). CoverNet ( Phan-Minh et al. (2020)) and MultiPath ( Chai et al. (2020)), which quantize the trajectory space to a set of anchors, often generate map-agnostic trajectories that cross non-drivable areas because the surrounding map is not considered. Recently, several studies have exploited the map information to obtain more performant goal candidates based on the assumption that vehicles follow lanes. TNT ( Zhao et al. (2021)) uses goal points sampled from a lane centerline, and GoalNet ( Zhang et al. (2021)) uses lane segments as trajectory anchors. However, while previous methods assume that the likelihood of arriving at a final destination is random, they assume that trajectories are unimodal in order to reach a specific goal area. In this paper, we assume inherent uncertainty in which trajectories can vary due to the interactions with surrounding vehicles in order to reach a specific goal area.

## 2.2 INTERACTION MODELING

Considering the interaction between agents helps to predict a socially aware trajectory. In the very early stage, interaction is obtained by pooling interaction features in the local region ( Deo & Trivedi (2018); Gupta et al. (2018)). In other works, researchers attempted to obtain interaction through attention-based ( Ngiam et al. (2022); Mercat et al. (2020); Vemula et al. (2018)) or GNN-based method ( Carrasco et al. (2021); Cao et al. (2021); Zeng et al. (2021); Casas et al. (2020); Liang et al. (2020); Gao et al. (2020)). However, in most previous methods, interactions between agents are learned only with regression loss, which is insufficient to represent dynamic and rapidly changing situations. There exists a line of works that employs Neural Relational Inference (NRI) ( Kipf et al. (2018)) that explicitly predicts and learns interaction using a latent interaction graph. EvolveGraph ( Li et al. (2020)) utilizes two interaction graphs, static and dynamic, and NRI-MPM ( Chen et al. (2021)) uses a relation interaction mechanism and spatio-temporal message passing mechanism. Similarly, we apply the NRI-based method to predict and train the interaction explicitly.

## 2.3 MULTI-MODAL TRAJECTORY PREDICTION

Trajectory prediction is a stochastic problem, which means that there are multiple possible futures instead of a unique answer. Recently, deep generative models like GAN (Goodfellow et al. (2014)) or VAE (Kingma & Welling (2013)) have been employed to address this issue. GAN-based (Gupta et al. (2018); Kosaraju et al. (2019); Li et al. (2021b)) and VAE-based models (Ivanovic & Pavone (2019); Salzmann et al. (2020); Tang & Salakhutdinov (2019)) predict multiple futures by sampling multiple latent vectors. A well-organized latent space is necessary to sample meaningful latent vectors for predicting diverse, yet plausible future trajectories. This has become a natural choice in recent works ( Ma et al. (2021); Bae et al. (2022)). The work most closely related to ours is GRIN (Li et al. (2021a)), which argues that multi-modality in trajectory prediction comes from two sources: personal intention and social relations with other agents. However, GRIN only considers past interaction, while we propose to consider future interaction by taking into account the characteristics of vehicle motion. Since vehicle motion mainly follows lanes, we utilize lane information to infer future interactions.

## 3 FORMULATION

In each scene, the past and future trajectories of $N$ vehicles are observed. The past trajectory $\mathbf{x}_t^-$ consists of positions for $-t_p : 0$ timesteps before the current timestep, and the future trajectory $\mathbf{x}_t^+$ consists of positions for $1 : t_f$ timesteps after the current timestep ($t = 0$). Lane information is obtained from the HD map, which consists of $M$ segmented lane polylines. The lane information is represented as a graph: $\mathcal{G} = (\boldsymbol{\ell}, \mathbf{e})$, where the nodes ($\boldsymbol{\ell}$) correspond to the different lane segments, and the edges ($\mathbf{e}$) represent the relationships between the segments. There are five relationship between segments: *predecessor*, *successor*, *left/right neighbor* and *in-same-intersection*. The input to the model is denoted as $\mathbf{X}$, which consists of the past and future trajectories of the vehicles and the lane information. Here, the future trajectories is only used in training. The output of the model is denoted as $\mathbf{Y}$, which consists of $F$ predicted future trajectories for each agent. The model also predicts the future lane occupancy (i.e., which vehicles are occupying which lanes) as a medium using a probability distribution $\boldsymbol{\tau}_t$ for each vehicle and lane segment at each future timestep: $1 : t_f$. The predicted future lane occupancy is denoted as $\boldsymbol{\tau}_t^-$, and the ground truth future lane occupancy is denoted as $\boldsymbol{\tau}_t^+$.

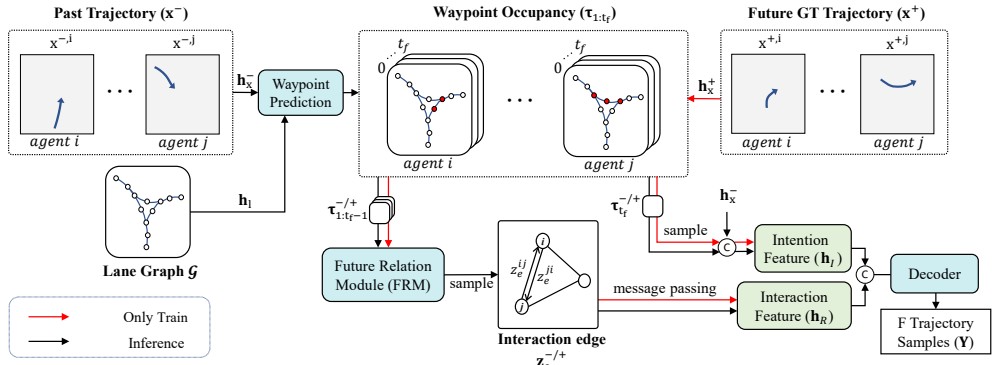

Figure 4: Overall structure of the proposed method. Given past/future motion inputs, the waypoint occupancy ($\boldsymbol{\tau}_{1:t_f}$) is obtained. The goal features are then sampled following $\boldsymbol{\tau}_{t_f}$. Intention feature is derived from the goal features and the past motion ($h_x^-$). The Future Relationship Module (FRM) utilizes the intermediate waypoint occupancy ($\boldsymbol{\tau}_{1:t_f-1}$) to sample the interaction edges among agents. Message passing is then performed to obtain the interaction feature. Finally, the decoder predicts $F$ future trajectories from concatenation of intention and interaction features.

## 4 METHOD

Our focus is on modeling the "Future Relationship" between agents. A naive method to infer the Future Relationship is to predict all future vehicle trajectories and then calculate similarity among them. However, this method is inefficient and redundant, as it requires performing prediction twice. Moreover, the criteria for calculating similarity between trajectories may not be clear. In this paper, we utilize lane information for modeling the Future Relationship, inspired by the idea that the vehicles mainly follow lanes. Our key idea is that *if two vehicles are expected to pass on adjacent lanes, they will have a high chance of interacting in the future*.

We present the overall structure of our method in Fig. 4. First, we predict the waypoint occupancy, which represents the probability of a vehicle passing a specific lane segment during future time steps $1 : t_f$ (Sec. 4.1). Using this information, our Future Relationship Module (FRM) infers interaction as an edge feature connecting agent node pairs (Sec. 4.2). These interaction edges are used to transfer information between agent nodes to form the interaction feature through message passing. Finally, in the decoding stage (Sec. 4.3), the decoder predicts future trajectories from the aggregation of the interaction feature and intention feature, which is derived from the concatenation of past motion and goal features. Following AgentFormer (Yuan et al. (2021)), our method is based on CVAEs where the condition corresponds to the intention, and the latent code corresponds to the interaction feature. Then, we compute prior and posterior distribution of the interaction feature, as described in Sec. 4.2.2, 4.2.3.

### 4.1 WAYPOINT OCCUPANCY

In this section, we describe how to obtain the waypoint occupancy. We need two waypoint occupancies: one predicted from past trajectory ($\boldsymbol{\tau}_t^-$) and the ground truth ($\boldsymbol{\tau}_t^+$) for obtaining prior and posterior distributions of interaction, respectively.

To predict the waypoint occupancy from past trajectory, we first encode the past trajectories $\mathbf{x}^-$ and the lane graph $\mathcal{G}$ into past motion and lane features: $\boldsymbol{h}_x^-, \boldsymbol{h}_\ell$. Then, following TNT (Zhao et al. (2021)), we predict the waypoint occupancy as Eq. (1). Here, $[\cdot, \cdot]$ denotes concatenation, and we apply softmax to ensure that the waypoint occupancy sum up to one: $\sum^M \tau_t^m = 1$.

$$\boldsymbol{\tau}_{1:t_f}^- = \text{softmax}(\text{MLP}([\boldsymbol{h}_x^-, \boldsymbol{h}_\ell])) \quad \in \mathbb{R}^{N \times M \times t_f} \tag{1}$$

For GT waypoint occupancy, we can directly obtain it from GT future trajectory since we know the position and heading of vehicles. More details can be found in the supplementary material.

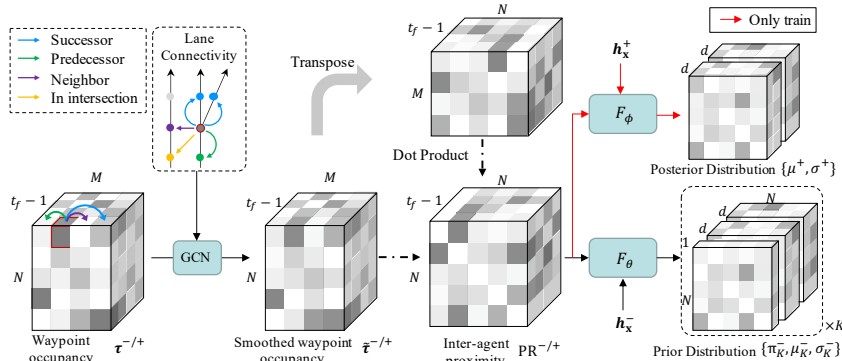

Figure 5: Future Relationship Module. During inference, predicted waypoint occupancy ($\tau^-$) is fed to GCN, dot-producted by itself to obtain inter-agent proximity (PR$^-$). Prior of interaction ($\pi_K$, $\mu_K, \sigma_K$) is then obtained as Gaussign Mixture. During training, GT waypoint occupancy ($\tau^+$) is fed to obtain posterior of interaction ($\mu, \sigma$) as Gaussian distribution.

## 4.2 FUTURE RELATIONSHIP MODULE (FRM)

Fig. 5 shows the FRM, which consists of three parts: computing inter-agent proximity and obtaining posterior and prior distribution. From intermediate waypoint occupancy of vehicle ($\tau_{1:t_f-1}$), we compute how each pair of vehicle pass adjacent lanes adjacent to each other at each timestep (inter-agent proximity). Based on that information and agents' past motion features, we obtain two distribution of interaction. In the following sections, we describe the details of each part.

### 4.2.1 INTER-AGENT PROXIMITY

To compute the inter-agent proximity (PR), we first smooth the waypoint occupancy using a Graph Convolutional Network (GCN) ( Welling & Kipf (2016)). The reason for doing so is that when a vehicle passes a specific lane, it affects other vehicles that pass the adjacent lane, not necessarily the same lane. Therefore, we apply different smoothing for each lane connectivity (predecessor, successor, neighbor, in-the-same-intersection) by employing 2-hop GCN layers. Specifically, each layer aggregated information from neighboring lanes and applies a non-linear transformation. This allows the model to capture spatial dependencies among agents and improve the accuracy of the inter-agent proximity computation. Each layer is expressed as Eq.(2) where $\sigma$, $D_e$, $A_e$ and $W_e$ are softmax followed by ReLU, degree, adjacency and weight matrix for each edge type, respectively.

$$\tilde{\boldsymbol{\tau}}_{1:t_f-1} = \sum_{\substack{e \in \{succ, pred, \\ right, left, inter\}}} \sigma \left( D_e^{-1} A_e \boldsymbol{\tau}_{1:t_f-1} W_e \right) \quad \in \mathbb{R}^{N \times M \times (t_f-1)} \quad (2)$$

With this smoothed waypoint occupancy, we can compute the inter-agent proximity using the dot product of $\tilde{\boldsymbol{\tau}}_{1:t_f-1}$ across the lane axis.

$$\text{PR} = \tilde{\boldsymbol{\tau}}_{1:t_f-1} \cdot \left( \tilde{\boldsymbol{\tau}}_{1:t_f-1} \right)^\top \quad \in \mathbb{R}^{N \times N \times (t_f-1)} \quad (3)$$

### 4.2.2 PRIOR OF THE INTERACTION

To obtain the prior distribution, we use the past motion features ($\boldsymbol{h}_{\text{x}}^-$) and inter-agent proximity. In this subsection, we omit superscript - for simplification. There are two design factors for our interaction modeling: (i) interaction should reflect diverse and stochastic properties, and (ii) it occurs in every pair of vehicles. Consequently, the prior distribution is defined as Gaussian Mixture (GM) per agent pair. Then, we define interaction edge $e^{ij}$ between agent i and j as a d-dimensional feature ($p_\theta(\boldsymbol{e}|\mathbf{X}) \sim \prod_{k=1}^K \boldsymbol{\pi}_k \mathcal{N}(\boldsymbol{\mu}_k, \mathbf{I}\sigma_k^2)$) following GMVAE ( Dilokthanakul et al. (2016)). The distribution parameters ($\boldsymbol{\mu}_K, \boldsymbol{\sigma}_K, \boldsymbol{\pi}_K = \{\boldsymbol{\mu}_K^{ij}, \boldsymbol{\sigma}_K^{ij}, \boldsymbol{\pi}_K^{ij}\}^{1:N,1:N}$) are obtained from the neural network $F_\theta$:

$$\boldsymbol{\mu}_K^{ij}, \boldsymbol{\sigma}_K^{ij}, \boldsymbol{\pi}_K^{ij} = F_\theta([pr^{ij}, h_{\text{x}}^i, h_{\text{x}}^j]) \quad \in \mathbb{R}^{Kd}, \mathbb{R}^{Kd}, \mathbb{R}^K \quad (4)$$

$F_\theta$ is composed of MLP layers and 1-d conv layer (Deo & Trivedi (2018)). We then perform two sampling steps, one for the interaction mode $k$ (from $\boldsymbol{\pi}_K$) and one for $\epsilon$ (from Gaussian noise). This allows for K distinct interactions modes:

$$\boldsymbol{\mu}_k, \boldsymbol{\sigma}_k = \text{argmax}_k(\boldsymbol{\pi}_K + \boldsymbol{g}), \quad \boldsymbol{g} \sim \text{Gumbel}(0, 1) \tag{5}$$

$$\boldsymbol{z}_e^- = \boldsymbol{\mu}_k + \boldsymbol{\sigma}_k \boldsymbol{\epsilon} \quad \in \mathbb{R}^{N \times N \times d}, \quad \boldsymbol{\epsilon} \sim \mathcal{N}(\mathbf{0}, \mathbf{I}) \tag{6}$$

Next, we compute the interaction feature ($\boldsymbol{h}_R^- = \{h_R^i\}^{1:N}$) via message passing from sampled interaction edge, as follows:

$$h_R^i = \sigma'(\frac{1}{N-1}\sum_{j \neq i}^{N} z_e^{ij} \otimes F_p(h_{\text{x}}^j)) \quad \in \mathbb{R}^d \tag{7}$$

### 4.2.3 CVAE POSTERIOR

To obtain the posterior distribution, we use GT waypoint occupancy ($\boldsymbol{\tau}^+$) and the future motion feature ($\boldsymbol{h}_{\text{x}}^+$), which is obtained from GT future trajectory and same motion encoder with past trajectory. Similarly, we omit superscript + in this subsection. Inter-agent proximity is obtained with same procedure in Eqs. (2)-(3). The difference from the prior is that the posterior is modeled in a single Gaussian ($\boldsymbol{\mu}, \boldsymbol{\sigma} = \{\mu^{ij}, \sigma^{ij}\}^{1:N,1:N}$). Thus, $F_\theta$ is replaced with $F_\phi$:

$$\mu^{ij}, \sigma^{ij} = F_\phi([pr^{ij}, h_{\text{x}}^i, h_{\text{x}}^j]) \quad \in \mathbb{R}^d, \mathbb{R}^d \tag{8}$$

Then we sample $\boldsymbol{\epsilon} \sim \mathcal{N}(\mathbf{0}, \mathbf{I})$, and interaction edge is obtained: $\boldsymbol{z}_e^+ = \boldsymbol{\mu} + \boldsymbol{\sigma}\boldsymbol{\epsilon}$. Finally, following Eq. (7), interaction features ($\boldsymbol{h}_R^+$) is obtained.

### 4.3 DECODER

The decoder predicts future trajectories from the aggregation of the interaction feature ($\boldsymbol{h}_R$) and intention feature ($\boldsymbol{h}_I$). Here, the intention feature is obtained from past motion feature and goal feature following TNT. During training, the unique GT intention feature is repeated $F$ times, and we sample the interaction feature ($\boldsymbol{h}_R^+$) $F$ times from the posterior distribution. During inference, the intention feature is obtained from the past motion feature ($\boldsymbol{h}_{\text{x}}^-$) and goal feature ($\boldsymbol{h}_g^-$), which is sampled $F$ times from predicted waypoint occupancy at the final timestep ($\boldsymbol{\tau}_{t_f}^-$). The interaction feature ($\boldsymbol{h}_R^-$) is sampled $F$ times from the prior distribution. The decoder is composed of 2-layer MLP and predicts sequence of x,y coordinates. More details can be found in the supplementary material.

### 4.4 TRAINING

Because the GT waypoint occupancy ($\boldsymbol{\tau}^+$) is available, we can train the model to predict waypoint occupancy ($\boldsymbol{\tau}^-$) using negative log-likelihood (NLL): $\mathcal{L}_{nll} = -\boldsymbol{\tau}^+ log(\boldsymbol{\tau}^-)$.

However, since the interaction edge $\boldsymbol{z}_e$ is unobservable, we optimize the evidence lower bound (ELBO) to train the interaction distribution using the CVAE scheme.

$$ELBO = -\mathbb{E}_{q_\phi}[log(p_\theta(\mathbf{Y} \mid \mathbf{X}, \boldsymbol{z}_e, \boldsymbol{\tau})] + KL[q_\phi(\boldsymbol{z}_e \mid \mathbf{X}, \boldsymbol{\tau}) \parallel p_\theta(\boldsymbol{z}_e \mid \mathbf{X}, \boldsymbol{\tau})] \tag{9}$$

Here, $q_\phi$ is the approximate posterior, and $p_\theta$ is the prior. Since our model only allows the posterior to be Gaussian distribution, we can simplify the Kullback–Leibler (KL) divergence term as follow:

$$\mathcal{L}_{KL} = -KL[q_\phi \parallel p_\theta] \approx log\sum_k \boldsymbol{\pi}_k \exp(-KL[q_\phi \parallel p_{\theta,k}]) \tag{10}$$

The detailed derivation with the reparameterization trick can be found in the supplementary material. However, a common drawback with the NRI-based method is the "degenerate" issue, where the decoder tends to ignore the relation edge during training. To address this issue, we train the network to give different roles to the intention and interaction features. Since the GT trajectory is conditioned on the GT goal feature, we use the GT goal feature to compute the reconstruction term. This training strategy restricts the role of interaction edge to momentary motion, resulting in the following reconstruction loss: $\mathcal{L}_{recon} = \min_{\boldsymbol{z}_e}\{\mathbb{E}[log(p_\theta(\mathbf{Y} \mid \mathbf{X}, \boldsymbol{z}_e, \boldsymbol{\tau}^+)]\}$.

Finally, the overall loss is the sum of the three losses, which are trained jointly: $\mathcal{L}_{all} = \mathcal{L}_{nll} + \mathcal{L}_{KL} + \mathcal{L}_{recon}$.

Table 1: Comparison on nuScenes test set. Best in **bold**, second best in underline.

| Paper | mADE$_5$ | mADE$_{10}$ | MR$_5$ | MR$_{10}$ | mFDE$_1$ |
|---|---|---|---|---|---|
| Trajectron++ Salzmann et al. (2020) | 1.88 | 1.51 | 0.70 | 0.57 | 9.52 |
| P2T Deo & Trivedi (2020) | 1.45 | 1.16 | 0.64 | 0.46 | 10.5 |
| AgentFormer Yuan et al. (2021) | 1.86 | 1.45 | - | - | - |
| LaPred Kim et al. (2021) | 1.47 | 1.12 | 0.53 | 0.46 | 8.37 |
| MultiPath Chai et al. (2020) | 1.44 | 1.14 | - | - | 7.69 |
| GOHOME Gilles et al. (2022a) | 1.42 | 1.15 | 0.57 | 0.47 | 6.99 |
| Autobot Girgis et al. (2021) | 1.37 | 1.03 | 0.62 | 0.44 | 8.19 |
| THOMAS Gilles et al. (2022b) | 1.33 | 1.04 | 0.55 | 0.42 | 6.71 |
| PGP Deo et al. (2022) | 1.27 | 0.94 | 0.52 | 0.34 | 7.17 |
| Ours | **1.18** | **0.88** | **0.48** | **0.30** | **6.59** |

Table 2: Comparison on Argoverse val/test set. Best in **bold**, second best in underline.

| Paper | Val set | | Test set | |
|---|---|---|---|---|
| | mADE$_6$ | mFDE$_6$ | mADE$_6$ | mFDE$_6$ |
| TNT Zhao et al. (2021) | 0.73 | 1.29 | 0.94 | 1.54 |
| LaneRCNN Zeng et al. (2021) | 0.77 | 1.19 | 0.90 | 1.45 |
| TPCN Ye et al. (2021) | 0.73 | 1.15 | 0.87 | 1.38 |
| Autobot Girgis et al. (2021) | 0.73 | 1.10 | 0.89 | 1.41 |
| mmTransformer Liu et al. (2021) | 0.72 | 1.21 | 0.84 | 1.34 |
| SceneTransformer Varadarajan et al. (2022) | - | - | 0.80 | 1.23 |
| Multipath++ Varadarajan et al. (2022) | - | - | 0.79 | 1.21 |
| HiVT Zhou et al. (2022) | 0.66 | 0.96 | **0.77** | **1.17** |
| Baseline | 0.71 | 1.03 | 0.86 | 1.30 |
| Ours | 0.68 | 0.99 | 0.82 | 1.27 |

# 5 EXPERIMENTS

We train and evaluate our method on two popular real-world trajectory datasets: nuScenes ( Caesar et al. (2020)) and Argoverse ( Chang et al. (2019)). nuScenes/Argoverse datasets provide the 2/2 seconds of past and require 6/3 seconds of future trajectory at 0.5/0.1 second intervals, respectively. Training/validation/test sets consist of real-world driving scenes of 32,186/8,560/9,041 in nuScenes and 205,942/39,472/78,143 in Argoverse. For the baseline model in ablation, we follow TNT for goal conditioned model, and MHA encodes interaction from past trajectories. For implementation and computation details, please refer to the supplementary material.

## 5.1 QUANTITATIVE RESULT

Our method outperforms SoTA models in all nuScenes benchmark metrics, as shown in Tab. 1. Specifically, our model outperforms the runner-up method, PGP ( Deo et al. (2022)), by a substantial margin. This result indicates that our explicit interaction modeling via inferring waypoint occupancy helps scene understanding compared to the implicit interaction modeling of PGP. When predicting 10 samples, our model shows improvements of 5.3% and 8.8% in terms of mADE and MR. Previously, THOMAS ( Gilles et al. (2022b)) was ranked first in mFDE$_1$ by proposing a recombination module that post-processes marginal predictions into the joint predictions that are aware of other agents. However, our model performs better than THOMAS in mFDE$_1$, indicating better interaction modeling ability without post-processing. This is possible because inferring future relationships helps to better understand the future interaction with other agents; details are provided in the ablation study.

We also evaluated our method on the Argoverse dataset. While our model does not achieve SoTA performance, it still shows remarkable performance improvement in both validation and test sets. Moreover, except for the HiVT, our method make competitive performance in mADE. Please note that our model (0.82) is still comparable to SceneTransformer (0.80) and Multipath++ (0.79) in the test set results. However, HiVT uses the surrounding vehicles' trajectories for training, resulting in increased training data. Therefore, a direct comparison to HiVT would be rather unfair.

We do not achieve SoTA in Argoverse because the proposed method is less effective than in nuScenes. We attribute this disparity to the differences in dataset configurations, where nuScenes requires predicting a longer future trajectory than Argoverse. As intuition suggests, interaction modeling has a more significant impact on longer-range prediction tasks. To validate this assumption, we conducted an ablation study by measuring the performance gain on nuScenes when predicting the same length of future as Argoverse. The results, presented in Tab. 3, shows that our interaction modeling method improves mADE$_1$ by over 10% in a 6-second prediction task, but its effect was halved in a 3-second prediction task, which is similar to the results obtained in Argoverse. This finding suggests that our interaction modeling method is more effective in longer-range prediction tasks.

Table 3: Impact of prediction time to the proposed modeling in terms of $\text{mADE}_1/\text{mADE}_6$.

|  | Baseline | Ours | Improvement |
|---|---|---|---|
| nuScenes (6sec) | 3.23/1.17 | 2.89/1.10 | **10.5%/6.0%** |
| nuScenes (3sec) | 1.26/0.50 | 1.19/0.48 | 5.6%/4.0% |
| Argoverse (3sec) | 1.41/0.71 | 1.33/0.68 | 5.7%/4.2% |

Table 4: Ablation studies on nuScenes.

|  | F=1 | F=5 |
|---|---|---|
|  | mADE/mFDE | mADE/mFDE |
| **Impact of model design** | | |
| Baseline | 3.23/7.60 | 1.26/2.49 |
| Ours w/o FR | 3.21/7.59 | 1.26/2.50 |
| Ours w/o GCN | 3.04/6.94 | 1.22/2.41 |
| Ours w/ Sym | 2.99/6.78 | 1.22/2.35 |
| **Importance of multimodal stochastic interaction** | | |
| Ours w/ GP | 2.98/6.78 | 1.20/2.33 |
| Ours w/ Deterministic | 2.96/6.80 | 1.28/2.52 |
| Ours (Full) | **2.89/6.61** | **1.19/2.30** |

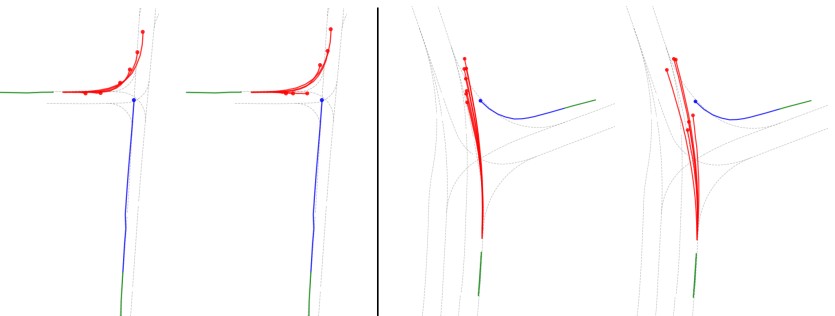

Figure 6: Qualitative results of the proposed method. The green solid line is past trajectories, the red lines are 6 predicted samples by baseline (left) and our method (right). The blue line is GT future trajectory of the surrounding vehicles. Lane centerlines are in gray dashed lines. In complex road scenes, baseline generates spatially uniform samples regardless of interaction with surrounding vehicles. On the other hand, our method generates diverse yet interaction-aware samples: wait or surpass other vehicles that would join in the future.

## 5.2 QUALITATIVE RESULT

In Fig. 6, we present prediction samples (F=6) from the baseline (left) and our method (right). To assess the efficacy of our method, we brought the samples with two agents and plotted the prediction of a single agent per scene. The green, blue, and red solid lines indicate the past trajectories of the both agents, future trajectories of the surrounding agents, and prediction samples of the target agents, respectively. In two scenes, each target agent sets its intention to where the other agent is likely to pass in the future. Our method generates prediction samples that incorporate and leverage the Future Relationship with other agents. Which means, unlike the baseline method that ignores other agents and generates spatially uniform trajectories, our model surpasses or waits for the other agents accounting for interaction. Moreover, not only considering two modes of interaction; surpass or wait, we also allow stochasticity within a single mode of interaction. Consequently, our model generates diverse yet interaction-aware samples.

Furthermore, our method can incorporate stochastic interaction when multiple agents are present. In the experiment shown in Fig. 7, we predict the trajectories of the target agents (denoted as 0) with multiple interacting vehicles. In each scene, the intention of the target vehicle is fixed (denoted in green) and two interaction edges are sampled. The corresponding predicted trajectory samples and the degree of interaction ($\|\boldsymbol{z}_e\|$) are plotted on the right. In the first row of the figure, the target agent (0) infers significant interaction with agent 2 in sample 1. As agent 2 is moving in the same direction and is predicted to move ahead, our model generates an accelerating trajectory to follow agent 2. In contrast, in sample 2, the interaction with agent 1 is sampled as significant because they are expected to be in the same intersection. In this case, our model generates decelerating trajectory considering the future motion of agent 1. Importantly, all predicted trajectories in these samples are appropriately constrained within the goal lane segments as the intention is set to the green colored lane. This indicates that our training strategy effectively restricts the role of interaction features to momentary motion.

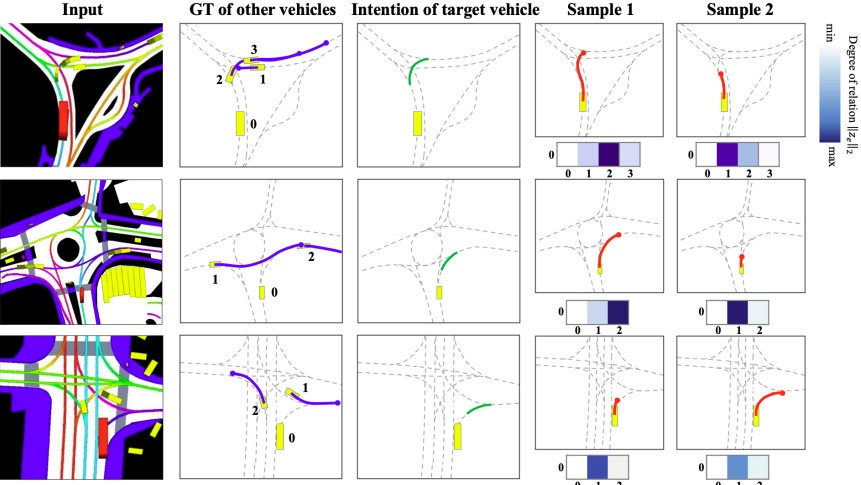

Figure 7: Qualitative results of the proposed method in multi-agent scene.

## 5.3 ABLATION STUDY

Ablation on the impact of model design is shown in the upper part in Tab. 4. The **Ours w/o FR** variant does not consider Future Relationship in interaction modeling, and only uses past trajectories to infer the relation, similar to the NRI. This variant performs almost identically to the baseline, which uses MHA of past trajectories to model interaction. This result shows the importance of leveraging Future Relationship for plausible interaction inference. The **Ours w/o GCN** variant omits the smoothing waypoint occupancy leading to inaccurate inter-agent proximity (PR) estimation, especially in the posterior distribution. Since GT waypoint occupancy is a binary value, computing PR from it can result in inaccurate proximity and lower prediction performance. In the proposed model, we allow asymmetric interaction between two agents. The **Ours w/ Sym** applies hard symmetric interaction modeling ( Li et al. (2019)), and it shows that our asymmetric design is more suitable for modeling the driver relation.

The importance of multi-modal stochastic interaction modeling is shown in the lower part of Tab. 4. The **Ours w/ GP** variant models the prior distribution as Gaussian distribution instead of GM, considering only a single modality of interaction, which leads to a performance decline compared to the full model with multi-modal interaction. The **Ours w/ Deterministic** variant predicts only the mean of interaction edges in Eq. 4. Although it can model multi-modal interaction, the diversity is prone to be limited compared to the stochastic counterpart especially when the sample size F is large. The result shows that stochastic modeling is critical for prediction performance, and deterministic modeling significantly degrades the prediction performance when predicting more samples. In contrast, the **Ours w/ GP** variant shows relatively less performance drop as it maintains stochasticity even after removing the GM prior.

## 6 CONCLUSION

In this paper, we propose Future Relationship to effectively learn the interaction between vehicles for trajectory prediction. By explicitly utilizing lane information in addition to past trajectories, our FRM can infer proper interactions even in complex road structures. The proposed model generates diverse yet socially plausible trajectory samples by obtaining interaction probabilistically, which provides explainable medium such as waypoint occupancy or inter-agent proximity. We trained our model using CVAE scheme and validated it on popular real-world trajectory prediction datasets. Our approach achieved SoTA performance in a long-range prediction task, nuScnes, and brings remarkable performance improvement in a short-range prediction task, Argoverse. Modeling Future Relationship is a novel approach, and we anticipate that using more sophisticated training methods (Ye et al. (2022); Zhou et al. (2022)) or a better baseline model (such as GANet (Wang et al. (2022)) may further improve prediction performance.

ACKNOWLEDGMENTS

This work was supported by Institute of Information & Communications Technology Planning & Evaluation(IITP) grant funded by the Korea government(MSIT) (No.2014-3-00123, Development of High Performance Visual BigData Discovery Platform for Large-Scale Realtime Data Analysis.

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
