# OpenReview forum: "Leveraging Future Relationship Reasoning for Vehicle Trajectory Prediction"
_ICLR.cc/2023/Conference — ICLR 2023 poster_

### Official Review · Reviewer_2v4y · 2022-10-19

**Confidence:** 4
**Correctness:** 3
**Technical Novelty And Significance:** 3
**Empirical Novelty And Significance:** 3
**Recommendation:** 6

**Clarity, Quality, Novelty And Reproducibility:**

The paper's math notation is not clean and the authors should simplify them.

The proposed future relationship module has clear methodological contribution to the community.

The paper is quite dense and the model contains multiple sub-components (e.g. waypoint occupancy, GMVAE, FRM, etc). Without the code from authors, it is not easy to reproduce the results.

**Strength And Weaknesses:**

Strength:

1. Compared with previous methods which directly use the history to predict the future, this method add an intermediate step, which first predict future waypoint occupancy on lane segments, and then derive explicit future relationships between agents. The paper show that such explicit future relationships are very helpful for predicting the final trajectory of an agent, and the model achieve SoTA results on nuScenes dataset.

2. To derive future relationship, the authors proposed to calculate inter-agent pair-wise feature vectors that follow Gaussian Mixture distributions, and ablation study shows that modeling the stochastic of this feature vector is important.

Therefore, this paper has good methodological contribution. Meanwhile, the experiments are thorough with quantitative, qualitative and ablation results.

Weaknesses:
1. The authors should really work on simplifying the notations. There are too many sup- and subscript and it is hard to follow.

2. Why is the waypoint occupancy conditioned on the sampled goal lane features? Can we integrate out the goal? The Gaussian Mixture  in FRM should be able to capture the stochasticity of goals. This will reduce inference time.

3. In section 4.4, the authors says "the goal feature (\hat{h}^{-}_{\ell})" is sampled F times from the inferred waypoint occupancy, however, in section 4.1 the waypoint occupancy is derived from the goal feature. The two statements seem contradictory.

4. z_{e}^{ij} in equation (7) already contains information about h_x^j, why do we need another element-wise product with h_x^j? Does experiment show that this is necessary?

5. Why is the Argoverse test set results not included in Table 3?

6. The method does introduce quite a lot of computation compared with other baselines such as MultiPath++. It would be better if the authors can report training and inference cost compared with baselines.

**Summary Of The Paper:**

The authors proposed to leverage future relationships for vehicle trajectory prediction. Particularly, the method first predicts future waypoint occupancy on lane segments. Then, a future relationship module defines an interaction edge between every two agents based on the proximity and features of the two agents, and the interaction edges follow Gaussian mixture distributions. Conditioned on the interaction features and intention features, the decoder predicts the trajectory. The results is SoTA on nuScenes dataset and on par to SoTA on Argoverse dataset.

**Summary Of The Review:**

The paper proposed to use vehicles' future relationships to improve trajectory prediction. The paper has clear science contribution and has achieved SoTA results on public dataset.

---

> ### Author Response · Authors · 2022-11-12
> **Response to Reviewer 2v4y (1)**
>
> We thank reviewer 2v4y for providing valuable questions and thoughtful comments to improve the paper.
> We especially appreciate the reviewer for noting the inefficiency in waypoint prediction. We were able to improve our method based on the comments in the revised paper as described below.
>
> **Notation Simplification.**
>
> We simplified the notation in Sec.3 in the revised version.
> Moreover, we omitted the superscript -/+ in Sec.4.2 for simplification.
>
> **Conditioning waypoint occupancy to goal lane feature.**
>
> We sincerely appreciate the insightful review and suggestion.
> In the initial experiment design, waypoint occupancy was conditioned on sampled goal lane features because we assumed that drivers set their goals first and then set intermediate waypoints.
> However, as the reviewer mentioned, this sequential goal-waypoint prediction method is inefficient.
> We thus tried the single pass waypoint prediction method following the reviewer’s comments.
> We verified that the single waypoint prediction also performs well, even with marginal performance gains in the nuScenes result.
> The changed contents can be found in the first paragraph in Sec.4.1 in the revised version.
> Moreover, the training time is reduced by about 2/3s for convergence.
> Most importantly, because GM can handle stochasticity with the single waypoint occupancy, its performance does not degrade.
> In addition, as training waypoint occupancy is more effective using single-pass prediction, the prediction performance increased slightly.
> We thank the reviewer again for providing valuable insight.
>
> **Contradicting representation.**
>
> There were typos in Sec.4.4 in the initial manuscript.
> The sentence “the goal feature is sampled F times from waypoint occupancy” should have been corrected to “the goal feature is sampled F times from goal probability.”
> There was miswriting because the goal probability is confused with the last timestep’s waypoint occupancy ($\tau_{t_f}$).
> In the revised paper, we corrected Sec.4.1 and Sec.4.4 according to our aforementioned response.
>
> **Additional use of $\pmb{h}_x^j$.**
>
> As the reviewer pointed out, $z_e$ already contains information of the past motion feature.
> However, we make another use of $h_\textup{x}$ to prevent losing motion information.
> Unfortunately, $z_e$ also contains random noise $\epsilon$, which makes the only use of this feature unstable.
> Therefore, we designed the interaction as a variant of the attention mechanism.
> Similar to the attention, the additional use of $h_x^j$ corresponds to “value” in the attention.
> The difference from the attention mechanism is that different types of relationships can be considered because we predict d-dimensional relation features while attention uses a single scalar value.
> In addition, our method has advantages in that it can also model stochastic interaction.
>
> **Computation.**
>
> We compared the computation cost by looking at the total FLOP count.
> For the Argoverse dataset, we estimated the total FLOP count through the computing capability of the GPU used and the training time in both SceneTransformer and Autobot:
> SceneTransformer takes 108,000,000 TFLOPs (420 TFLOPS TPU-v3 X 73 hours), while Autobot takes 396,000 TFLOPs (11 TFLOPS Nvidia 1080ti X 10 hours).
> Our model, on the other hand, takes 6,739,200 TFLOPs (39 TFLOPS Nvidia A6000 X 48 hours).
> Ultimately, our method performs competitively with SceneTransformer but with much less FLOPs, while shows outperforming performance compared to Autobot albeit with more FLOPs.
>
> **Reference**
>
> [C1] Wang, et al. "GANet: Goal Area Network for Motion Forecasting." arXiv (2022).

---

> ### Author Response · Authors · 2022-11-12
> **Response to Reviewer 2v4y (2)**
>
> **The test set result of Argoverse.**
>
> We did not report the test set results because we wanted to emphasize the performance improvement more than SotA performance.
> Following the comment, we included the result for the test set of Argoverse in Tab.1 below (as well as in the revised version).
> There are three published methods that outperform our method in the test set (SceneTransformer, Multipath++, HiVT).
> Since our method is more effective for long-range prediction as mentioned in Sec.5.1, it might be less effective in the Argoverse set, which focuses more on short-range prediction.
> However, please note that our model (0.82) is still comparable to SceneTransformer (0.80) and Multipath++ (0.79) in mADE.
> On the other hand, HiVT shows better performance (0.77) than other methods.
> It might be because, unlike other methods that learn only from the target vehicle trajectory, HiVT utilizes the surrounding agent’s trajectory for training.
> It results in increasing the training data, so a direct comparison to HiVT is rather unfair.
> In addition, despite this performance drop, our model still has strengths in terms of diversity and plausibility.
> Please refer to the qualitative comparison with HiVT included in Fig.1 of the revised supplementary material.
> There is a clear advantage to autonomous driving systems, where various risks must be taken into account.
>
> Furthermore, we expect that using more sophisticated training methods or a better baseline model (such as GANet [C1]) will also lead to performance improvements in our model.
>
> * [Table 1]: Comparison with other methods on the Argoverse valid/test set in $\textup{mADE}_6$.
> | Method | Val set | Test set |
> | :--- | :---: | :---: |
> | TNT | 0.73 | 0.94 |
> | DenseTNT | 0.73 | 0.88 |
> | LaneRCNN | 0.77 | 0.90 |
> | TPCN | 0.73 | 0.87 |
> | Autobot | 0.73 | 0.89 |
> | mmTransformer | 0.72 | 0.84 |
> | SceneTransformer | - | 0.80 |
> | Multipath++ | - | 0.79 |
> | HiVT | 0.66 | 0.77 |
> | Ours-baseline | 0.71 | 0.86 |
> | Ours-full | 0.68 | 0.82 |

---

### Official Review · Reviewer_Y1Xx · 2022-10-24

**Confidence:** 5
**Correctness:** 3
**Technical Novelty And Significance:** 2
**Empirical Novelty And Significance:** 2
**Recommendation:** 3

**Clarity, Quality, Novelty And Reproducibility:**

Details are in [Weaknesses]

Clarity:
The clarity is good but with some minor mistakes.

Quality:
Clear-written.

Novelty:
Novelty is not significant.

Reproducibility:
The training details are in the supplementary material, and authors promise to release the code after paper acceptance.

**Strength And Weaknesses:**

**Strengths**:
1. Well-organized and clearly written.
2. Promising results.


**Weaknesses**:
1.  Some details are missing. What is the temporal probability of passing the same lane? and how to learn interaction based on that probability? This sentence is confusing and ambiguous.


2. Novelty is limited. Almost all the techniques are well-studied in the vehicle trajectory prediction task, i.e., the motivation of using HD map (lane information), occupancy via probability, many prior works follow this approach now. The differences between the proposed method and existing baselines are not highlighted, nor the contributions are emphasized.


3. Minor mistakes:
(1) In Fig. 4, I believe it is Future Trajectory Samples instead of F Trajectory Samples.

**Summary Of The Paper:**

In this paper, authors utilize the lane information to model the future relationship among agents for trajectory prediction. Specifically, the probability of lane-level waypoint occupancy of vehicles is predicted, and a future relation module is developed for sampling interaction edges. Finally, interaction features are obtained via message passing for decoding. Empirical results on nuScenes and Argoverse are promising.

**Summary Of The Review:**

I list my concerns in [Weaknesses].

---

> ### Comment · Reviewer_6gEW · 2022-11-05
> **Can you expand on your lack of novelty complaint?**
>
> Regarding Weakness #2:
>
> I know it's common for competitive trajectory prediction models to process vector maps with GNNs, but this article proposed a very specific module to do feature encoding based on the strong assumption that agents will follow their current lane, and it appears to be (at least) competitive with other SotA methods. Can you specifically point to some prior art that you feel is too similar to the proposed method to merit novelty in their approach?

---

> > ### Comment · Reviewer_Y1Xx · 2022-11-05
> > **Most Concern**
> >
> > Hi Reviewer 6gEW and Authors:
> >
> > Sorry for the confusion. I was also one of the reviewers of this paper when it was submitted to NeurIPS'22. I checked both versions, the clarity improves a lot, the missing details are inclueded, and experiments are conducted on Argoverse.
> >
> > However, the entie contributions, title, and story are total different in these two versions but with the same models and the same results. The previous version was about long/short-term uncertainties/intentions, and this version focus on lane/map information. Thus it makes me somehow confused and unconvincing.
> >
> > I hope authors could provide some comments on this and I will increase my rating if it makes sense.

---

> > > ### Author Response · Authors · 2022-11-05
> > > **Differences from the NeurIPS 2022 submission.**
> > >
> > > Dear Reviewer 6gEW and Reviewer Y1Xx,
> > >
> > > First of all, we appreciate your efforts for a valuable discussion.
> > > As Reviewer Y1Xx mentioned, the proposed model in this paper is almost the same as the model in the NeurIPS 2022 submission.
> > >
> > > In the NeurIPS version, we tackled the trajectory prediction problem with the long/short-term intention formulation, and the short-term intention in the NeurIPS version corresponds to the interaction feature (obtained from future relationships) in the ICLR version.
> > > The long-term intention is briefly described in the ICLR version because, based on the NeurIPS reviews, we concluded that the novelty of the long-term intention is not significant.
> > > (Long-term intention was about how to sample the goal feature.
> > > You can find the contents about the long-term intention in Sec.2.3, Sec.4.1, Figure 4, and so on.).
> > >
> > > On the other hand, we added more details and experiments about the future relationship (short-term intention) for clarity in the ICLR version, while we could not provide enough details in the NeurIPS version because of the page limit. It might be the reason why Reviewer Y1Xx asked for clarification. We will further discuss your other comments later, and we hope this can answer your concern.

---

> ### Author Response · Authors · 2022-11-12
> **Response to Reviewer Y1Xx**
>
> We thank reviewer Y1Xx for the time and effort they put into providing valuable questions and comments to improve the paper.
> Here, we provide the response to the comments and questions as follows:
>
> **Temporal probability of passing the same lane.**
>
> The concept of the “temporal probability of passing the same lane” is visualized in the middle part of Fig.5.
> To obtain the "temporal probability of passing the same lane", the waypoint occupancy of vehicles is utilized (denoted as $\pmb{\tau}_{1:t_f-1} \in \mathbb{R}^{N \times M \times (t_f-1)}$).
> The waypoint occupancy is a 3D tensor that represents the probabilities that vehicle $i$ ($\in \mathbb{N}^{N}$) would pass lane node $m$ ($\in \mathbb{N}^{M}$) at future intermediate time step $t$ ($\in \mathbb{N}^{t_f-1}$).
>
> We cross-product the waypoint occupancy along lane axis, so we get another 3D tensor, the inter-agent proximity (denoted as $PR \in \mathbb{R}^{N \times N \times (t_f-1)}$).
> The PR represents how likely agents i ($\in \mathbb{N}^{N}$) and j ($\in \mathbb{N}^{N}$) will pass the same lane at time step $t$ ($\in \mathbb{N}^{t_f-1}$).
> The obtained PR corresponds to the "temporal probability of passing the same lane" between agent pairs.
>
>
> **Learning interaction from the PR.**
>
> As mentioned in the last sentence of the fifth paragraph of Sec.1, we borrow the concept from Neural Relation Inference (NRI) [B1] to learn interaction.
> First, we define an interaction graph in which agents become nodes and relations between agents become edges.
> The edge is inferred from the feature of influencer (giving interaction), reactor (receiving interaction), and the PR between the two vehicles (Eq.8 in the paper; please refer to the appendix below the response).
> We then compute interaction feature from node and edge feature by message passing (Eq.7 in the paper).
> The obtained interaction feature is considered as a latent code in a conditional variational autoencoder (CVAE) scheme.
>
> Similar to CVAE, the latent code is jointly learned by two losses: reconstruction and KL divergence.
> For the first loss, we decode future trajectory from sampled latent code (interaction feature), then compute L2 loss with GT future trajectory.
> For the second loss, we compute KL divergence between prior and posterior distribution of latent code.
> We compute the posterior from GT future trajectory following same process with prior.
> In the posterior distribution, the pseudo-GT of the interaction is included, so we induce the prior distribution to learn from that pseudo-GT.
>
> Meanwhile, there are two major differences between our method and the conventional NRI.
> First, we consider the Future Relationship by explicitly utilizing agent-wise proximity (PR).
> Here, the definition of Future Relationship can be found in the third paragraph in Sec.1 of the paper.
> We show its benefit in the ablation study: \textbf{ours w/o FR}.
> When the Future Relationship is not considered, there is little performance gain compared to the baseline because the MHA module already handles the interaction of the past trajectories.
> The second difference is that we perform stochastic relation inference.
> Looking at Eq.8 in the paper, we do not predict the relation edge directly but predict distribution parameters, $\mu$ and $\sigma$, assuming that the relation follows the Gaussian distribution.
> This design comes from the fact that the drivers, highly chaotic systems, should consider stochastic interaction either, unlike deterministic systems such as particles dealt with in the NRI paper.
> We also show its benefit in the ablation study: \textbf{ours w/ Deterministic}.
>
>
> **Novelty.**
>
> As the reviewer mentioned, many SotA methods studied the way to use lane information.
> The most common way to use lane information is to obtain lane-to-agent features by permutation-invariant operation (VectorNet [B2], Autobot [B3], LaneRCNN [B4], etc.).
> Another alternative is to regard a lane node as the vehicle’s goal candidate (TNT [B5], DenseTNT [B6], GOHOME [B7], etc.).
> In our paper, we utilize \emph{waypoint occupancy as a rough future motion for learning interaction}. To the best of our knowledge, our work is the first to utilize waypoint occupancy in such a way.
> As a result, we achieved SotA performance in long-range prediction and brought remarkable improvement to short-range prediction.
>
> **Minor mistakes.**
>
> We appreciate the reviewer's comments to improve our paper.
> The mistakes in Fig.4 were adequately corrected in the revised paper.

---

> > ### Author Response · Authors · 2022-11-12
> > **Appendix for the comments**
> >
> > **Appendix**
> >
> > (Eq.8 in the paper): $\mu^{ij}, \sigma^{ij} = F_{\phi} ( [ pr^{ij}, h_\textup{x}^i, h_\textup{x}^j ] )  \in \mathbb{R}^{d}, \mathbb{R}^{d}$
> >
> > where $pr$ represent agent-wise proximity, $h_\textup{x}$ represent agent motion feature. Agent i is
> >
> > (Eq.7 in the paper): $h_R^i =  \sigma' ( \frac{1}{N-1}\sum_{j \neq i}^{N} z_e^{ij} \otimes F_p (h_\textup{x}^j) ) \quad \in \; \mathbb{R}^{d}$
> >
> > where $h_R$ represent interaction feature, $\sigma'$ is activation, $N$ is number of agents, $z_e$ is relation edge, and $F_p$ is MLP layer. Here, agent i is reactor, and agent j in influencer.
> >
> > **Reference**
> >
> > [B1] Kipf, et al. "Neural relational inference for interacting systems." ICML (2018). \
> > [B2] Gao, et al. "Vectornet: Encoding hd maps and agent dynamics from vectorized representation." CVPR (2020). \
> > [B3] Girgis, et al. "Latent Variable Sequential Set Transformers for Joint Multi-Agent Motion Prediction." ICLR (2021). \
> > [B4] Zeng, et al. "Lanercnn: Distributed representations for graph-centric motion forecasting." IROS (2021). \
> > [B5] Zhao, et al. "Tnt: Target-driven trajectory prediction." CoRL (2021). \
> > [B6] Gu, et al. "Densetnt: End-to-end trajectory prediction from dense goal sets." CVPR (2021). \
> > [B7] Gilles, et al. "Gohome: Graph-oriented heatmap output for future motion estimation." ICRA (2022).

---

### Official Review · Reviewer_6gEW · 2022-11-03

**Confidence:** 3
**Correctness:** 3
**Technical Novelty And Significance:** 4
**Empirical Novelty And Significance:** 3
**Recommendation:** 8

**Clarity, Quality, Novelty And Reproducibility:**

Key insight is interesting, is clearly translated into the design Future Relationship Module, and it performs well on popular public datasets, so I think the content of the paper is good.

The presentation needs work. For example, h^+_x is presented in Figure 4 at the top of page 3 but not defined until the bottom of page 6, in Section 4.4. Furthermore, the English is choppy and sometimes grammatically incorrect -- the content is still intelligible, but it's often mildly distracting.

To the authors, I would recommend:
1) read through where all your variables are defined relative to figures and make sure that there's no more than ~1 page between their use and definitions
2) using an easily accessible English-based LLM e.g. GPT-3 from OpenAI, one paragraph at a time paste in your existing text and ask it to edit the text for grammar and clarity; this should be an easy way to clean up the text with little work on your end.
3) highlight in the conclusion that your Future Relationship Module is compatible with other SotA trajectory prediction methods and future work of combining them together will likely result in an even stronger method!

**Strength And Weaknesses:**

Strengths: Good performance on popular public Trajectory Prediction benchmarks (NuScenes and Argoverse 1). The novel contribution, the Future Relationship Module, is shown to benefit the performance of the network. Furthermore, Future Relationships Module appears to provide useful features that are orthogonal to the modules of other competitive methods, opening up the possibility of future work that combines these methods together.

Weaknesses: Paper's actualized key insight is "agents usually follow lanes" but they do not do any analysis to show how often this assumption is violated. Due to the fact that the method performs well, presumably it's not very often, but it would be nice to know often it emerges in the train/val dataset and qualitatively what happens to the predictions under these conditions.

NuScenes results not fully SotA due to missing models (https://www.nuscenes.org/prediction?externalData=all&mapData=all&modalities=Any). DGCN_ST_LANE has significantly lower MinADE_5 (1.092) vs the proposed method (1.19 as reported in the paper), a gap larger than reported against the runner up PGP, and DGCN_ST_LANE was submitted on May 25th 2022, which is significantly before the Sept 28 2022 ICLR submission deadline and thus should be mentioned in the results table. However, note that the authors' proposed method *does* outperform DGCN_ST_LANE on MinADE_10 (0.89 in the paper vs 0.974) and so it does maintain some claim to SotA.

Argoverse 1 results were reported on the *validation* set, not the test set on the public leaderboard. This important to note because the proposed method is outperformed on the validation set by HiVT. HiVT performs significantly worse on the test set -- HiVT-128, the method described in Zhou et al., 2022, has significant hits to both mADE_6 (0.66 val vs 0.77 test) and mFDE_6 (0.96 val vs 1.17 test), and is outperformed on this test set by many methods such as DCMS (Ye et al., https://arxiv.org/abs/2204.05859, posted April 12th, 2022) Wayformer (Nayakanti et al., https://arxiv.org/abs/2207.05844, posted July 12, 2022). Certainly the transitive performance degradation from the Argoverse 1 validation set to test set of HiVT may not apply to the authors' method, but this does leave a lingering question of how well the proposed method actually stacks up relative to the state of the art.

**Summary Of The Paper:**

Problem: Vehicle Trajectory Prediction as commonly defined and evaluated.

Key insight: agents usually follow lanes, so explicitly use lane graph structure to find likely future agent-agent interactions.

Novel contributions: Future Relationship Module that consumes waypoint occupancy estimates + lane graph and produces a prior distribution of interactions conditioned on the historical trajectory, trained against a posterior distribution conditioned on the full trajectory for training. Ablations show that this module is the key to system performance, as without it performance deteriorates to baseline which only uses historical trajectories.

Evaluation: competitive / SotA performance on standard metrics for popular public Trajectory Prediction benchmarks (NuScenes and Argoverse 1)



**Summary Of The Review:**

Tackles the standard problem of trajectory prediction via a key insight that is actualized into a method that's competitive on popular public benchmarks. Likely will have an impact on the problem's community, as the actualized key insight forms a module that should be compatible with other SotA methods and will likely produce better performing future methods.

Conditioned on
1) expansion of the quantitative results section to include the additional methods mentioned (plus other methods deemed relevant from benchmark performance)
2) clean-up of the presentation + English

I would be happy to see this paper presented at ICLR 2023, and I think it would make a nice addition to the trajectory prediction community.

---

> ### Author Response · Authors · 2022-11-12
> **Response to Reviewer 6gEW (1)**
>
> We thank reviewer 6gEW for thoughtful and detailed reviews and for providing valuable comments to improve the paper.
> We also appreciate that the reviewer valued our key insight and how to translate it into the design of the proposed modules.
> Here, we provide the response to the comments as follows:
>
> **How often is the assumption violated?**
>
> We thank the reviewer for the insightful question.
> Indeed, we checked the percentage of cases that a vehicle does not follow lanes in the nuScenes trajectory dataset.
> At first, to find the lanes that vehicles follow, we filtered lane candidates whose directions are within +/- 30 degrees from the vehicles’ heading, then selected the lane candidate closest to the vehicle’s position.
> We examined all future timesteps of the vehicle trajectory.
> If the distance between the vehicle’s position and the closest lane is smaller than half the width of the lane (generally 1.5m), it can be assumed that the vehicle follows the specified lane.
> From our findings of the dataset, the average distance between the vehicle and the supposed following lane is 0.57m with the vehicle following the lane 94\% of the entire dataset.
> Only a small fraction of cases (i.e., U-turns, noisy segments from detection error) violate the assumption.
> As commented, we too agree that it will be interesting and meaningful to evaluate the prediction under those conditions and will conduct more analysis accordingly.
>
> **SotA in nuScenes.**
>
> We tried to find the paper of DGCN\_ST\_LANE for reference, but we could not find the paper.
> Its result was hidden in [eval.ai leaderboard](https://eval.ai/web/challenges/challenge-page/591/leaderboard/1659), and we could not find the paper even in arxiv.
> If we find any information about DGCN\_ST\_LANE, we will cite it and compare the performance.
>
> **Helpful comments to improve presentation.**
>
> We appreciate the valuable comments for improving the presentation.
> We believe that the manuscript can be much more convincing by reflecting on the reviewer's comments.
> As suggested, we made the space between the use of variables and their definitions less than 1 page in the revised paper.
> We also proofread the manuscript using professional editing services for grammar, clarity, and conciseness.
> In the last paragraph, we also highlighted that our method is compatible with other SotA trajectory prediction methods and can be improved more if combined with other state-of-the-art methods.
>
> **Reference**
>
> [A1] Varadarajan, et al. "Multipath++: Efficient information fusion and trajectory aggregation for behavior prediction." ICRA (2022).   \
> [A2] Ngiam, et al. "Scene Transformer: A unified architecture for predicting future trajectories of multiple agents." ICLR (2021). \
> [A3] Bahari, et al. "Vehicle trajectory prediction works, but not everywhere." CVPR (2022). \
> [A4] Nayakanti, et al. "Wayformer: Motion forecasting via simple \& efficient attention networks." arXiv (2022). \
> [A5] Ye, et al. "DCMS: Motion Forecasting with Dual Consistency and Multi-Pseudo-Target Supervision." arXiv (2022). \
> [A6] Wang, et al. "GANet: Goal Area Network for Motion Forecasting." arXiv (2022).

---

> ### Author Response · Authors · 2022-11-12
> **Response to Reviewer 6gEW (2)**
>
> **Argoverse test set result.**
>
> As suggested by the reviewer, we report our result for the Argoverse test set and compare it with those of other methods as in Tab.1 below, in terms of $\textup{mADE}_6$. Here, we also include more relevant methods (Multipath++ [A1], SceneTransformer [A2]) in comparison as the reviewer suggested. (this comparison can also be found in the revised paper)* .
> Since our method is more effective for long-range prediction as mentioned in Sec.5.1, it might be less effective in the Argoverse set, which focuses more on short-range prediction.
> However, please note that our model (0.82) is still comparable to SceneTransformer (0.80) and Multipath++ (0.79).
> On the other hand, HiVT shows better performance (0.77) than other methods.
> It might be because, unlike other methods that learn only from the target vehicle trajectory, HiVT utilizes the surrounding agent's trajectory for training, thus increasing the training data.
>
> We can see that the performance decline of our method from the val set to the test set is not less than that of HiVT.
> From our analysis, goal-based models seem to have a larger performance drop in the test set as shown in Tab.1: the average mADE drop of the goal-based models is 22.8\%, and that of other methods is 17.7\%.
> Considering that the performance drop generally comes from data distribution gap, it seems that the distribution gap is larger in map data than in motion data in the Argoverse.
> The performance degradation due to the map data distribution gap is reported in a recent paper [A3].
> However, it is also reported that the decline could be mitigated by the learning method proposed in that paper.
> Therefore, we think that our method could achieve better test results by adopting [A3].
>
> In addition, despite this performance drop, our model still has strengths in terms of diversity and plausibility thanks to the goal-based method.
> Please refer to the qualitative comparison with HiVT included in Fig.1 of the revised supplementary material.
> There is a clear advantage to autonomous driving systems, where various risks must be taken into account.
>
> * [Table 1]: Comparison with other methods on the Argoverse valid/test set in $\textup{mADE}_6$
> | Category | Method | Val set | Test set | Decline ratio (from val to test) |
> | :--- | :--- | :---: | :---: | :---: |
> | Goal-based | TNT | 0.73 | 0.94 | -28.7% |
> |  | DenseTNT | 0.73 | 0.88 | -20.5% |
> |  | Ours-baseline | 0.71 | 0.86 | -21.2% |
> |  | Ours-full | 0.68 | 0.82 | -20.6% |
> |  | avg |  |  | -22.8% |
> | Other | LaneRCNN | 0.77 | 0.90 | -14.4% |
> |  | TPCN | 0.73 | 0.87 | -19.2% |
> |  | Autobot | 0.73 | 0.89 | -21.9% |
> |  | mmTransformer | 0.72 | 0.84 | -16.6% |
> |  | SceneTransformer | - | 0.80 | - |
> |  | Multipath++ | - | 0.79 | - |
> |  | HiVT | 0.66 | 0.77 | -16.6% |
> |  | avg |  |  | -17.7% |
>
>
> We also compare our method with more methods including the methods presented in arxiv papers as in Tab.2.
> Actually, in the original submission, according to the 10$^{th}$ and 11$^{th}$ questions in the FAQ of [ICLR guideline](https://iclr.cc/Conferences/2023/ACGuide), performance comparison was conducted only for the papers published in peer-reviewed conferences or journals before July 28th this year, excluding arxiv papers. However, we agree with the review that we need to include more methods in comparison. So, we include more methods presented in arxiv papers as in Tab.2.
> To the best of our knowledge, there are three methods (from the arxiv papers) reporting the performance better than HiVT on the Argoverse leaderboard: Wayformer [A4], DCMS [A5], and GANet [A6]. So, we compare our method with them in Tab.2.
>
> Here, please note that the methodologies and the information used for the methods are different, so a direct comparison of performance might not be fair.
> For example, among the methods in Tab.2, DCMS mainly proposes training methods for increasing supervision while we focus on model architecture.
> In addition, Wayformer uses traffic light information that might significantly affect performance, while the proposed method does not utilize that information.
> We believe that the use of more sophisticated training methods or additional inputs will also lead to performance improvements in our model.
>
> Compared to GANet, our model does not perform better but is still comparable in terms of mADE.
> Moreover, because GANet proposes how to sample a better goal lane, it could be easily combined with our method and bring superior performance in mFDE too.
>
> * [Table 2]: Comparison with more methods including arxiv papers on the Argoverse test set.
> | Method | mADE | mFDE |
> | :--- | :---: | :---: |
> | DCMS | 0.77 | 1.14 |
> | Wayformer | 0.77 | 1.16 |
> | GANet | 0.81 | 1.16 |
> | Ours | 0.82 | 1.27 |
>
>
> \* : For TPCN, we found that the performance reported in the paper is quite different from the performance on the leaderboard. So, we used the performance reported in the paper for the comparison.

---

### Decision · Program_Chairs · 2023-01-20

**Decision:**

Accept: poster

**Justification For Why Not Higher Score:**

This was broderline, and the paper currently has many weaknesses, as pointed out above.

**Justification For Why Not Lower Score:**

My final decision is influenced mainly by the soundness and simplicity of the idea and the strong results on one very well-known dataset.

**Metareview: Summary, Strengths And Weaknesses:**

The paper proposes a new approach for vehicle trajectory prediction with multiple vehicles.  Its main insight is that explicitly modeling future interactions between vehicles is a good intermediate step for eventual trajectory prediction, and that this can be done quite effectively by assuming lane-following to identify these possible interactions. This translates into a working algorithm, and results on nuScenes show strong performance improvements over the many prior published works on this benchmark.

Some weaknesses in this paper are:
- Performance is only convincing on nuScenes, and the author reports on Argoverse test set results are somewhat poor, indicating that the method may rely on assumptions that don't generally hold. It is strongly recommended to the authors that they present these results in future iterations / camera-ready.
- Related to the above point, there isn't a clear analysis of weaknesses and failure cases. For example, in what cases, if any, does the lane-following assumption for interaction prediction fail, and does this then translate to trajectory prediction failures too?
- Computation concerns about the 2-phase trajectory prediction inference, raised during review phase, are yet to be fully addressed.
- Notation clutter in the original paper, and other writing weaknesses, were not addressed in the form of actual paper revisions during the response stage, but authors have promised to fix these.

We have other recommendations to the authors:
- They should more clearly distinguish from prior works like TNT / GoalNet that also use lane information, but in different contexts (not for future interaction prediction).

**Note From Pc:**

if the above contains the word "oral" or "spotlight" please see: "oral" presentation means -> notable-top-5% and "spotlight" means -> notable-top-25%. As stated in our emails, we are disassociating presentation type from AC recommendations

**Summary Of Ac-Reviewer Meeting:**

We used the AC-reviewer meeting for this paper to address divergences among reviewers, and to clearly identify what the objective facts that everyone agrees on are, and where exactly subjective disagreements begin. For example, everyone agrees that the performance on the first dataset, NuScenes, is very good, and that on the second dataset, Argoverse, is disappointing, and that there is no analysis of why. Reviewers had differing opinions on whether the technique is still interesting.